complexity/statistical physics

markets, transfer entropy, risk spillover, causality

**Author for correspondence:**
Tomaso Aste
e-mail: t.aste@ucl.ac.uk

# Causal coupling between European and UK markets triggered by announcements of monetary policy decisions

Vittoria Volta[1] and Tomaso Aste[1,2,3]

[1]Department of Computer Science, University College London, Gower Street, London, UK
[2]UCL Centre for Blockchain Technologies, UCL, London, UK
[3]Systemic Risk Centre, London School of Economics and Political Sciences, London, UK

TA, 0000-0002-4219-0215

We investigate high-frequency reactions in the Eurozone stock market and the UK stock market during the time period surrounding European Central Bank (ECB) and the Bank of England (BoE)'s interest rate decisions, assessing how these two markets react and co-move influencing each other. The effects are quantified by measuring linear and nonlinear transfer entropy combined with a bivariate empirical mode decomposition from a dataset of 1 min prices for the Euro Stoxx 50 and the FTSE 100 stock indices. We uncover that central banks' interest rate decisions induce an upsurge in intraday volatility that is more pronounced on ECB announcement days and there is a significant information flow between the markets with prevalent direction going from the market where the announcement is made towards the other.

## 1. Introduction

The policy frameworks within which central banks operate have been subject to major changes over the recent decade. Indeed, in response to the COVID-19 pandemic, central banks have significantly upgraded their financial stability functions, expanding on programmes first tried during the Global Financial Crisis (GFC) as well as developing an entirely new set of initiatives in order to promote credit flows and keep economies afloat. While the specific actions by each central bank were logically determined by the idiosyncrasies of their economies and institutions [1], there are several common threads: the reliance on a more multidimensional set of tools; the size, speed and breadth of the responses that had no precedent. While these programmes were crucial to stabilize economies and financial markets, this expansion of responsibilities—well beyond the narrow

inflation-targeting focus of most central banks—also raises numerous questions about the role of monetary policy and central banks in the future.

Not only central banks have gained more power on the financial markets, but also the progressively more interconnected global environment has influenced the degree to which cross-border asset prices react to monetary policy decisions. According to [2], capital flow volatility and the cross-border correlation of asset price movements and credit growth have increased in recent years, in connection with unconventional monetary policies put in place and with the intensifying search for a yield in a low interest rate environment. This has revived the debate over the risks posed by international spillovers of monetary policies across countries, not only to financial stability but also to monetary policy autonomy.

Given the considerable sway that central banks nowadays hold over the financial markets and the increasing global financial integration, the knowledge of how financial markets interact dynamically to the release of monetary policy decisions is of fundamental interest. It is a key input for policy makers who need to evaluate the impact of policy measures and monitor its transmission across different financial markets. It is also of particular benefit for market participants who are involved in intraday trading and need to assess the market risk of their positions induced by both foreign and domestic monetary policy announcements.

In this paper, we aim to explore how financial markets—specifically the Eurozone market and the UK market—react, co-move and might cause each other movements surrounding interest rate decisions made by the European Central Bank (ECB) and the Bank of England (BoE). The impact of central banks' announcements on stock market movements has been extensively covered in the literature. For instance, Kurihara [3] examines the impact of the ECB monetary policy announcements on stock prices and exchange rates in the Euro area; his findings indicate that most macroeconomic news shocks do not have any desirable impact on stock prices and exchange rates except the case of unemployment news on exchange rates. Among the earlier papers based on intraday observations, Farka [4] reports a significant effect of policy shocks on the level and volatility of US stock returns. Andersson [5] examines bond and stock market volatility reactions in the euro area and the US following their respective economies' monetary policy decisions. He found a strong upsurge in intraday volatility at the time of the release of the monetary policy announcements by the two central banks. Lunde and Zebedee [6] explore the impact of Federal Open Market Committee (FOMC) announcements on the intraday volatility dynamics of the S&P 500 index. Their analysis shows elevated intraday volatility through the market close, with a spike at the time of the announcement.

While the topic is certainly not new, to the best of our knowledge, this study represents one of the first attempts to investigate the dynamic interdependence between markets surrounding monetary policy decisions and quantify spillover effects in a high-frequency and non-parametrical way. This paper contributes to the existing literature in multiple aspects. From the content perspective, traditionally, papers studying the effects of monetary policy announcements on asset prices have mainly relied on the *level* effects on financial markets (see, for example, [7–10]). However, equity returns merely inform about an increase/decrease of an asset in reaction to announcements. We believe that more information could be gained by looking at the *volatility* which measures the markets' sensitivity—that is, whether markets are getting more nervous (volatility increase) or calmer (volatility decrease)—and therefore, it mirrors the immediate reaction on financial markets during an announcement day. This paper explicitly measures the linear and nonlinear interrelations between the European and UK stock markets volatilities surrounding ECB and BoE announcement days, which to our knowledge has not been addressed in earlier research.

This paper introduces a novel methodology to investigate potential interactions between the two stock markets volatilities by using a data-driven approach which consists of two steps.[1] First, a bivariate empirical mode decomposition (BEMD) technique is used to decompose the European and UK stock markets volatilities into several pairs of intrinsic mode functions (IMFs) and residual functions with different timescales. Specifically, we use the BEMD algorithm developed by [12], which is an extension of the traditional empirical mode decomposition (EMD) and is data-driven, hence, it has obvious advantages in processing non-stationary and nonlinear data [13]. Second, the linear and nonlinear transfer entropy are applied to inspect the causality[2] between the residues and the IMFs of

---

[1]A similar method has been previously used by [11] to investigate the relationship between China's stock market and the international oil market. However, our approach to detect nonlinear causality is based on information-theoretic measures rather than a non-parametric test statistics.

[2]In this paper, we consider a statistical form of causality, which can be observed in co-dependent time series where a response in the dependent series is more likely to follow after some change in the driving series. If the response of the dependent series scales as a

each pair. To investigate linear relations, we use a generalization of the Granger–Geweke causality test, which assumes linearity and employs vector auto-regressive techniques to detect the impact of past values of a variable $X_t$ on future values of another variable $Y_t$. In fact, it has been shown [14] that, for multivariate Gaussian variables, Granger-causality and transfer entropy are equivalent. To detect nonlinear relations, we adopt the measure formalized by Schreiber [15], known as transfer entropy, which offers a natural way to model statistical causality between variables in multivariate distributions. Specifically, it quantifies the reduction in uncertainty about the dependent time series provided by the past values of the driving signal, conditioning on the past values of the dependent variable itself [16]. This method is model-free, and as [17] point out in their analysis on networks of international stock market indices, is an effective way to quantify the time-directed transfer of information between stochastic variables in the nonlinear case.

Only a few papers to date have investigated the response of stock index prices to monetary policy actions using high-frequency data. One of the primary obstacles of working with intraday data has been the accessibility to high-quality, high-frequency prices over a relatively long calendar time period.[3] This paper contributes to the existing high-frequency empirical literature by examining a unique dataset of historical 1 min prices for the Euro Stoxx 50 stock index and the FTSE 100 stock index from December 2015 to December 2019. The use of intraday data allows for better isolation of the response of stock index prices to the minutes release, since no other economic news is systematically released within such a narrow (1 min) window around the monetary announcements.

This paper uncovers four main findings. First, both central banks' interest rate decisions induce an upsurge in intraday volatility; however, the reaction of the markets appears to be more pronounced following ECB decisions than following BoE decisions. Second, there is a clear information flow between markets surrounding the press releases on announcement days. Third, such information flow spans across timescales and it is statistically significant for the majority of the intrinsic mode functions. Fourth, we found that the nonlinear transfer entropy results are consistent with the linear ones and do not seem to add any extra element. We therefore believe the linear approach to transfer entropy is sufficient in detecting the spillover effects between the Eurozone stock market and the UK stock market.

## 2. Results

In order to investigate the impact of central banks' interest rate decisions on the stock markets, we examine the average realized volatility profile on announcement days and the excess volatility profile, measured as a ratio between the average volatility on announcement days over the average volatility on non-announcement days. A ratio above one can be interpreted as the monetary policy decisions inducing 'higher than normal' volatility [5]. These are reported in figure 1.

Three interesting features can be observed from figure 1. First, both ECB and BoE's interest rate decisions induce an upsurge in intraday volatility in their respective domestic and foreign markets (figure 1a,b). Such volatility appears to be 'higher than normal' on the stock markets surrounding the time of the press release (figure 1c,d), and is particularly pronounced on ECB announcement days, where the average excess volatility is nearly 4.5 on the Euro Stoxx 50 and 2.5 on the FTSE 100. Second, some volatility persistence can be observed on both markets and can be noted up to more than 1 h after the central banks' decisions have taken place (figure 1c,d). Third, the Introductory Statement read by the ECB President induces 'higher than normal' volatility on the euro area.

Any interpretation on the basis of figure 1 could be spurious if important macro announcements were systematically released on the same days and at the same time as the ECB and BoE's monetary policy decisions. Therefore, we collected a significant number of US and euro area macro announcements to examine whether the interest rate decisions coincide with the release time of these macro statistics. The results of this examination suggest that none of the announcements under consideration occurred at the same time as the European Central Bank and Bank of England decisions, therefore, we believe that the observed upsurge in volatility is determined by the actual decisions and does not reflect market reactions to other macro news. By contrast, the ECB press conference is usually held at times of important US macro announcements, making the excess volatility measure difficult to interpret [5].

---

linear multiple of the driving signal, the relationship is defined as a linear coupling. If, instead, the response follows some other functions of the driving series, the relationship is nonlinear.

[3]See, for example, [18,19].

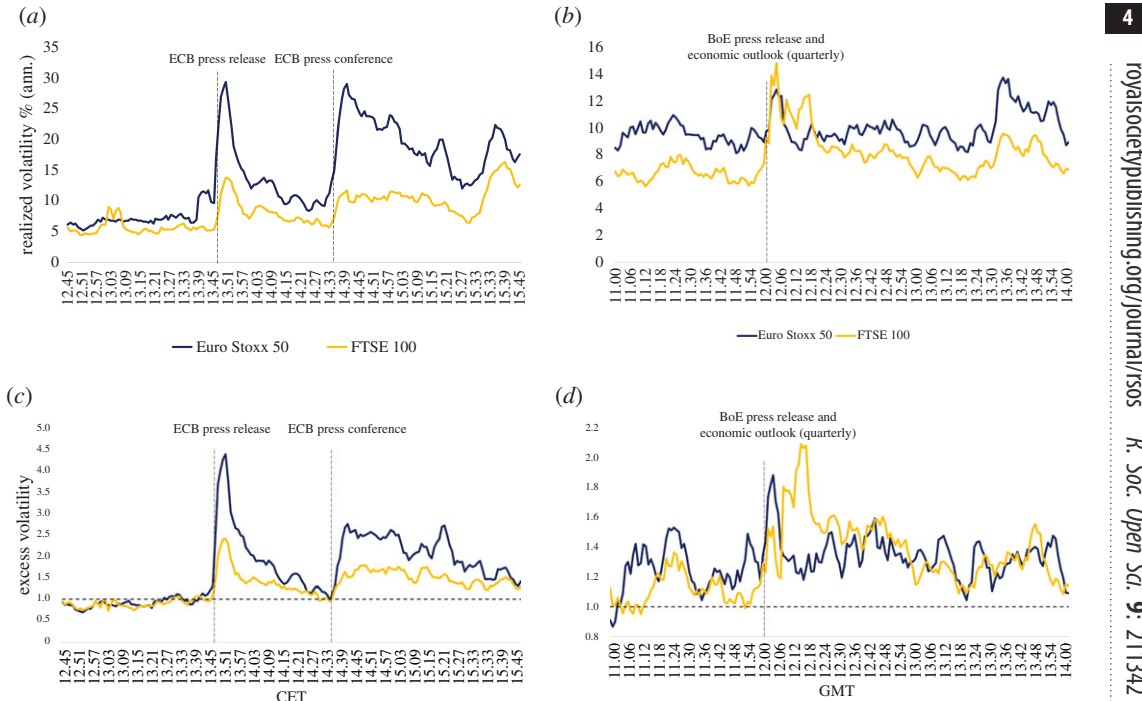

**Figure 1.** Average intraday volatility and average excess volatility on central banks' announcement days. The sub-figures above (*a,b*) display the Euro Stoxx 50 (blue line) and FTSE 100 (yellow line) average intraday volatility across a 3 h window on ECB and BoE announcement days. The volatility has been calculated using equation (4.2). The sub-figures below (*c,d*) show the average excess volatility, which is defined as the ratio between the average realized volatility on announcement days and the average realized volatility on non-announcement days. (*a*) Intraday volatility on ECB announcement days, (*b*) intraday volatility on BoE announcement days, (*c*) excess volatility on ECB announcement days and (*d*) excess volatility on BOE announcement days.

However, the ECB press conference is outside of the scope of this paper, which purely focuses on the market reaction to the release of central banks' interest rate decisions.

In order to minimize the impact that news releases may have on the statistical causality of the two stock market volatilities, we carry out the transfer entropy analysis on a 1 h window surrounding the interest rate announcements. Specifically, we select the time period 13.30 to 14.30 (CET) for ECB announcement days and the time period 11.45 to 12.45 (GMT) for BoE announcement days and adjust the time series such that they are both expressed in the same time zone. To investigate the interrelation between the Eurozone market and the UK market and examine potential spillover effects, we apply the methodology described in §4.2 to the average volatility time series. We decompose the Euro Stoxx 50 and the FTSE 100 average volatility series sampled around ECB and BoE announcement days and quantify linear and nonlinear causality relations both between the original signals and between the intrinsic mode functions (IMFs) originated from the bivariate decomposition.

Figure 2 shows the BEMD. From the signal decomposition, we get four pairs of IMFs and one pair of residual functions. We interpret the results in conjunction with table 1 which reports the average oscillating period for each BEMD component. The first mode function captures the volatility jumps that take place around the time of the press release, indicating that such fluctuations of the markets due to the release of monetary policy decisions are primarily short term. However, it is interesting to note that the high volatility in proximity to the announcements is partly transferred to lower-frequency components such as IMF2. This suggests that the volatility fluctuations driven by short-term factors spans between 1 to 4 min after the central banks' monetary policy decisions. The IMF3 instead captures the medium-term fluctuations of the volatility that ranges between 6 to approximately 8 min. Little information is instead conveyed by the last IMF and the residual. The IMF4, particularly on ECB announcement days, is on a large timescale of about 20–30 min and represents the long-term trends in the two markets.

The results of the linear and nonlinear causality investigation are shown in tables 2 and 3, where transfer entropy values, *p*-values, *z*-scores and information flow values are provided. In general, it is expected that causal links should be strongest at time-lags closest to the true signal lag, and gradually

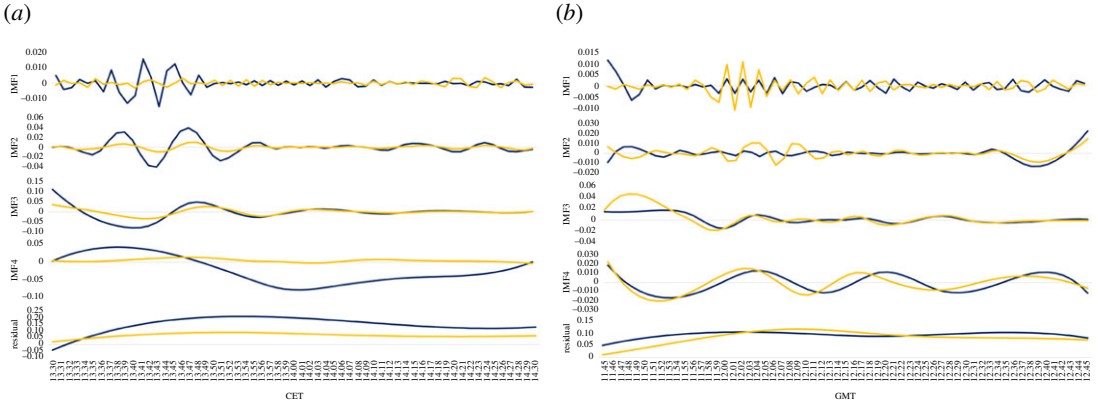

(*a*)                    (*b*)

**Figure 2.** Decomposition of average realized volatilities via BEMD. The average realized volatilities on ECB and BoE announcement days are used to create complex signals, where the real part (blue line) is represented by the Euro Stoxx 50 average volatility and the imaginary part (yellow line) corresponds to the FTSE 100 average volatility. The complex value signals are then decomposed into IMFs and residual functions, according to the bivariate empirical mode decomposition approach described in §4.2. (*a*) Volatility decomposition on ECB announcement days, (*b*) volatility decomposition on BoE announcement days.

**Table 1.** Average oscillating periods (minutes). This table shows the average oscillating periods for the Euro Stoxx 50 volatility series and the FTSE 100 volatility series measured in minutes for the extracted modes. The residue is a constant or a monotonic slope, therefore, it is the non-oscillating drift of the data.

| | timescale (minutes) | | | |
| | ECB announcement days | | BoE announcement days | |
| mode | Euro Stoxx 50 | FTSE 100 | Euro Stoxx 50 | FTSE 100 |
|---|---|---|---|---|
| IMF1 | 1.48 | 1.32 | 1.45 | 1.29 |
| IMF2 | 3.38 | 3.58 | 2.65 | 2.77 |
| IMF3 | 7.62 | 7.62 | 7.62 | 6.77 |
| IMF4 | 30.50 | 20.33 | 8.71 | 8.71 |
| residue | — | — | — | — |

decay as the time-lag considered is increased [16]. Therefore, in order to correctly identify causality, we perform the transfer entropy analysis over lags from 1 to 10, and pick the time-lag corresponding to the highest $z$-score.

Based on the data in tables 2 and 3, the following observations can be made regarding ECB announcement days. The original signal and both short-term and medium-term intrinsic mode functions show a consistent, clear direction of information flow from the Euro Stoxx 50 to the FTSE 100, suggesting that there is a spillover of volatility that is transferred from the Eurozone market to the UK market on ECB announcement days. With respect to the long-term scale (residual function), linear and nonlinear transfer entropy reveal opposite results. However, in absolute terms, we can observe that there is a greater net information transfer from the Euro Stoxx 50 to the FTSE 100 (0.16) in table 2, than from the FTSE 100 to the Euro Stoxx 50 (0.08) in table 3. The significance of these results can be validated with the $z$-score and $p$-value. At the significance level of $\alpha = 5\%$, the linear transfer entropies are significant in both directions, for the original signal and all extracted modes. The nonlinear transfer entropy results instead are statistically significant only in one direction for the majority of the intrinsic mode functions. This seems to suggest that the nonlinear approach to transfer entropy is less sensitive in detecting statistical causality between the two volatility time series on ECB announcement days.

With respect to BoE announcement days, we report the results in tables 4 and 5. We observe that the linear net transfer entropy (information flow) is positive for the original signals and all extracted modes, indicating a linear spillover effect from the FTSE 100 to the Euro Stoxx 50 on BoE announcement days. The direction of the nonlinear information flow is instead less obvious, with IMF1 and residual function that show opposite directions. However, similarly to tables 2 and 3, it can be noted that for IMF1 and

**Table 2.** Multi-scale linear transfer entropy surrounding ECB announcement days between 13.30 and 14.30 CET. This table shows the linear transfer entropy, $p$-value, $z$-score for the original signal and all the extracted modes on ECB announcement days. The variable $X$ denotes the Euro Stoxx 50 average realized volatility, whereas the variable $Y$ indicates the FTSE 100 average realized volatility. The information flow is the net transfer entropy, precisely the difference between $TE_{X \to Y}$ and $TE_{Y \to X}$. In the validation column, the symbol '$\to$' indicates unidirectional statistical significance, '$\leftrightarrow$' denotes a bidirectional statistical significance and '$*$' means no statistical significance. The significance level is 5%.

| mode | Lag | $TE_{X \to Y}$ | $p$ | $z$ | $TE_{Y \to X}$ | $p$ | $z$ | Inf. flow | | validation |
| --- | --- | --- | --- | --- | --- | --- | --- | --- | --- | --- |
| original | 2 | 0.34 | 0.00 | 26.31 | 0.18 | 0.00 | 13.12 | 0.16 | $\to$ | Euro Stoxx $\leftrightarrow$ FTSE |
| IMF1 | 1 | 0.08 | 0.00 | 5.90 | 0.05 | 0.01 | 3.46 | 0.02 | $\to$ | Euro Stoxx $\leftrightarrow$ FTSE |
| IMF2 | 1 | 0.35 | 0.00 | 29.13 | 0.24 | 0.00 | 17.09 | 0.10 | $\to$ | Euro Stoxx $\leftrightarrow$ FTSE |
| IMF3 | 5 | 0.72 | 0.00 | 53.38 | 0.46 | 0.00 | 31.83 | 0.25 | $\to$ | Euro Stoxx $\leftrightarrow$ FTSE |
| IMF4 | 10 | 0.49 | 0.00 | 24.50 | 0.39 | 0.00 | 18.41 | 0.10 | $\to$ | Euro Stoxx $\leftrightarrow$ FTSE |
| residual | 1 | 0.89 | 0.00 | 70.96 | 0.73 | 0.00 | 55.27 | 0.16 | $\to$ | Euro Stoxx $\leftrightarrow$ FTSE |

**Table 3.** Multi-scale nonlinear transfer entropy surrounding ECB announcement days between 13:30 and 14:30 CET. Notation and symbols are the same as in table 2.

| mode | Lag | $TE_{X \to Y}$ | $p$ | $z$ | $TE_{Y \to X}$ | $p$ | $z$ | Inf. flow | | validation |
| --- | --- | --- | --- | --- | --- | --- | --- | --- | --- | --- |
| original | 3 | 0.23 | 0.00 | 7.70 | 0.07 | 0.04 | 2.01 | 0.15 | $\to$ | Euro Stoxx $\leftrightarrow$ FTSE |
| IMF1 | 1 | 0.08 | 0.03 | 2.43 | 0.00 | 0.79 | -0.79 | 0.08 | $\to$ | Euro Stoxx $\to$ FTSE |
| IMF2 | 1 | 0.27 | 0.00 | 9.28 | 0.08 | 0.04 | 2.17 | 0.19 | $\to$ | Euro Stoxx $\leftrightarrow$ FTSE |
| IMF3 | 2 | 0.22 | 0.00 | 7.28 | 0.02 | 0.34 | 0.06 | 0.19 | $\to$ | Euro Stoxx $\to$ FTSE |
| IMF4 | 7 | 0.12 | 0.02 | 2.74 | 0.08 | 0.08 | 1.41 | 0.04 | $\to$ | Euro Stoxx $\to$ FTSE |
| residual | 10 | 0.09 | 0.10 | 1.32 | 0.17 | 0.01 | 3.51 | −0.08 | $\leftarrow$ | Euro Stoxx $\leftarrow$ FTSE |

residual function, the nonlinear net transfer entropy in table 5 is smaller in absolute values than the linear net transfer entropy in table 4. Moreover, except for IMF2, the linear transfer entropy values are significant in both directions, while the nonlinear transfer entropy values is significant only in the direction from FTSE 100 to the Euro Stoxx 50 for the first intrinsic mode function and instead in both directions for the second. Similarly to ECB announcement days, we conclude that the linear approach to transfer entropy performs better than the nonlinear approach in capturing the statistical causality between the volatility time series on BoE announcement days.

Before drawing final conclusions on the results for the announcement days it is important to investigate the statistical causality between the Euro Stoxx 50 volatility and the FTSE 100 volatility in the same time windows but on non-announcement days. The results are reported in tables S1–S4 in electronic supplementary material. We observe that, on days in which interest rates announcements are not released, there is a lack of consistency in the direction of information flow across the IMFs for both the linear and nonlinear case. Moreover, the nonlinear transfer entropy results appear to be non-statistically significant for some of the intrinsic mode functions (electronic supplementary material, tables S2 and S4), revealing a certain degree of disconnection between the Eurozone and the UK stock markets. In our view, this is strong evidence that the announcements have significant effects on the markets and their causal coupling. The lack of an obvious direction of causality on non-announcement days is instead an indication that the interaction between the two markets is more complex and hidden and unstable when no external driving factors are at play.

## 3. Discussion

In this paper, we use a high-frequency, data-driven approach to investigate how the Euro Stoxx 50 and the FTSE 100 stock market volatilities react, co-move and influence each other surrounding central banks' interest rate decisions. According to the above analysis, several interesting conclusions have emerged.

**Table 4.** Multi-scale linear transfer entropy surrounding BoE announcement days between 11.45 and 12.45 GMT. This table shows the linear transfer entropy, p-value, z-score for the original signal and all the extracted modes on BoE announcement days. The variable X denotes the FTSE 100 average realized volatility, whereas the variable Y indicates the Euro Stoxx 50 average realized volatility. The information flow is the net transfer entropy, precisely the difference between $TE_{X \to Y}$ and $TE_{Y \to X}$. In the validation column, the symbol '→' indicates unidirectional statistical significance, '↔' denotes a bidirectional statistical significance and '*' means no statistical significance. The significance level is 5%.

| mode | Lag | $TE_{X \to Y}$ | p | z | $TE_{Y \to X}$ | p | z | Inf. flow | | validation |
|---|---|---|---|---|---|---|---|---|---|---|
| original | 9 | 0.17 | 0.00 | 8.53 | 0.10 | 0.00 | 5.38 | 0.06 | → | Euro Stoxx ↔ FTSE |
| IMF1 | 2 | 0.16 | 0.00 | 11.35 | 0.05 | 0.02 | 3.15 | 0.11 | → | Euro Stoxx ↔ FTSE |
| IMF2 | 1 | 0.04 | 0.01 | 3.36 | 0.03 | 0.06 | 1.63 | 0.01 | → | Euro Stoxx → FTSE |
| IMF3 | 3 | 0.45 | 0.00 | 32.19 | 0.11 | 0.00 | 7.51 | 0.33 | → | Euro Stoxx ↔ FTSE |
| IMF4 | 3 | 0.44 | 0.00 | 31.78 | 0.34 | 0.00 | 24.53 | 0.09 | → | Euro Stoxx ↔ FTSE |
| residual | 1 | 0.18 | 0.00 | 14.33 | 0.07 | 0.00 | 4.76 | 0.10 | → | Euro Stoxx ↔ FTSE |

**Table 5.** Multi-scale nonlinear transfer entropy surrounding BoE announcement days between 11.45 and 12.45 GMT. Notation and symbols are the same as in table 4.

| mode | Lag | $TE_{X \to Y}$ | p | z | $TE_{Y \to X}$ | p | z | Inf. flow | | validation |
|---|---|---|---|---|---|---|---|---|---|---|
| original | 9 | 0.19 | 0.00 | 4.39 | 0.01 | 0.72 | −0.70 | 0.18 | → | Euro Stoxx → FTSE |
| IMF1 | 2 | 0.11 | 0.00 | 3.79 | 0.14 | 0.00 | 4.60 | −0.03 | ← | Euro Stoxx ↔ FTSE |
| IMF2 | 9 | 0.04 | 0.28 | 0.32 | 0.02 | 0.53 | −0.38 | 0.02 | → | Euro Stoxx * FTSE |
| IMF3 | 1 | 0.22 | 0.00 | 7.57 | 0.00 | 0.75 | −0.75 | 0.21 | → | Euro Stoxx → FTSE |
| IMF4 | 3 | 0.40 | 0.00 | 12.05 | 0.13 | 0.00 | 3.82 | 0.26 | → | Euro Stoxx ↔ FTSE |
| residual | 10 | 0.12 | 0.04 | 2.23 | 0.18 | 0.00 | 3.71 | −0.06 | ← | Euro Stoxx ↔ FTSE |

First, both the ECB and the BoE decisions induce an upsurge in intraday volatility. The reaction on the markets following the ECB's decisions are more pronounced compared with the reaction following BoE's decisions. We believe the two best possible explanations are as follows. Firstly, over the past 10 years, the Bank of England has shown much greater flexibility and willingness to pursue quantitative easing, increase the money supply and cut interest rates where necessary. Indeed, as it is shown in table S5 in the electronic supplementary material, BoE has cut interest rates twice as often as ECB did.[4] Such accommodating monetary policy where central banks are trapped in a 'do whatever it takes' mindset, gripping markets tighter rather than releasing control, may inadvertently prolong a low volatility environment. Secondly, while the European Central Bank is responsible for the inflation rate in the whole of Europe, the Bank of England has to consider the inflation only in the UK economy. Therefore, while the ECB's actual monetary policy decisions may be hard to predict, one could expect that BoE's decisions can be better anticipated by market participants. As the surprise effect following BoE's decisions is lower, also the volatility spike and the spillover effect on other markets may be less pronounced.

Second, there is a clear direction of information flow surrounding central banks' interest rate decisions. Specifically, the spillover of volatility is transferred from the Eurozone market to the UK market on ECB announcement days and from the UK market to the Eurozone market on BoE announcement days. Despite a very few exceptions, these directions are consistent across the majority of the intrinsic mode functions at different timescales and are detected by both the linear and nonlinear methodologies. Such a clear directionality is instead not detected for days in which no interest rate decisions are released from central banks.

Third, the spillover effects surrounding central banks' interest rate releases are robust, consistent and statistically significant. Therefore, the FTSE 100 market pricing may be affected by the Euro Stoxx 50 market on ECB announcement days and vice versa on BoE announcement days.

---

[4]This is partly attributable to their inflation target: BoE has a target of CPI = 2% + / −1, whereas the ECB has an inflation target of below, but close to, 2%. Therefore, the ECB is less tolerant of inflation going above the inflation target.

Fourth, the majority of the nonlinear transfer entropy results turned out to be statistically significant only in one direction. Therefore, in this context, the nonlinear approach is less efficient in detecting the spillover effects between the volatility time series, and the linear approach is instead sufficient in investigating the causality between the Eurozone stock market and the UK stock market. This indicates that the kind of coupling between the markets is prevalently linear.

From the above conclusions, some implications for the related stakeholders and market regulators can be derived. For investors, central banks' interest rate release affects pragmatic investment and risk mitigation decisions. Monetary policy announcements are often a catalyst for big directional stock movements, either up or down. Market participants could eliminate the investment risk and take advantage of big moves in the markets by using derivatives products such as options to design hedging strategies. For the market regulators, since the interest rate decisions have a significant impact on the asset prices, they need to be concerned about a potential shock from the stock markets and formulate responding strategies.

Building on this study, a key direction for future research would be to find further evidence of spillover effects following interest rate decisions by the Federal Reserve. It has been widely reported that the asset price sensitivity to US news is stronger compared with euro area news, partly owing to the fact that the US is considered among investors as the main engine of global growth [5]. However, little is still known on the volatility transmission of Fed decisions across foreign markets. Given that a break in trading may mitigate the impact of spillover effects, we believe that more information could be gained by carrying out the analysis on the futures markets, which operate nearly 24 h a day.

# 4. Material and methods

## 4.1. Data

Our data consist of 1 min intraday prices for the Euro Stoxx 50 stock market index and the FTSE 100 stock market index from December 2015 to December 2019. The data is from FirstRate Data.[5] From the original dataset, we select the following fields: Date, Time (recorded in local time zone) and Close Price. To study the statistical causality surrounding the European Central Bank and Bank of England's monetary policy decisions, we further reduce the original dataset by selecting only trading days in which monetary policy announcements were made. Both central banks' interest rate decisions are typically released on Thursday every six weeks. Hence, the size of our sample amounts to 32 and 35 trading days for ECB and BoE announcement days, respectively, with 510 prices per trading day.

Data on ECB and BoE's monetary policy decisions have been collected internally from the historical archives of the official ECB and BoE websites and from Trading Economics, a provider of an economic calendar, time series statistics, business news, long-term forecasts and short-term predictions.[6] As regards the timing, the BoE's interest rate decisions are usually released at 12.00 (GMT), and the ECB's interest rate decisions at 13.45 (CET). It should be noted that BoE's interest decisions, are also accompanied by a quarterly Monetary Policy Report that sets out the economic analysis and inflation projections that the Monetary Policy Committee (MPC) uses to make its interest rate decisions. This implies that, particularly for the Bank of England, there are two potential sources of new information released at the same time. In contrast to the BoE, the ECB's interest rate decisions and statements are announced to the public at separate times. The actual outcome of monetary policy decisions are released at 13.45 (CET), however, details about the economic and monetary analyses underlying each interest rate decision are instead conveyed in the Introductory Statement read by the ECB President 45 min later.

For each day $t$, we compute the continuously compounded intraday returns, $r_{t+i\Delta}$, using the following equation, which leads to 509 returns per trading day

$$r_{t+i\Delta} = \ln(p_{t+i\Delta}) - \ln(p_{t+(i-1)\Delta}), \tag{4.1}$$

where $p$ denotes the price and $\Delta$ is the sampling interval which in this case is 1 min. In order to minimize the impact that other news releases may have on the statistical causality between the two indices, we select a window size of one hour surrounding the monetary policy announcements, hence, our analysis is performed over 60 returns per trading day. Specifically, we select the time period 13.30 to

[5]The original dataset can be found at the homepage: https://firstratedata.com/.

[6]The web page of the economic calendar can be found at: https://tradingeconomics.com/calendar.

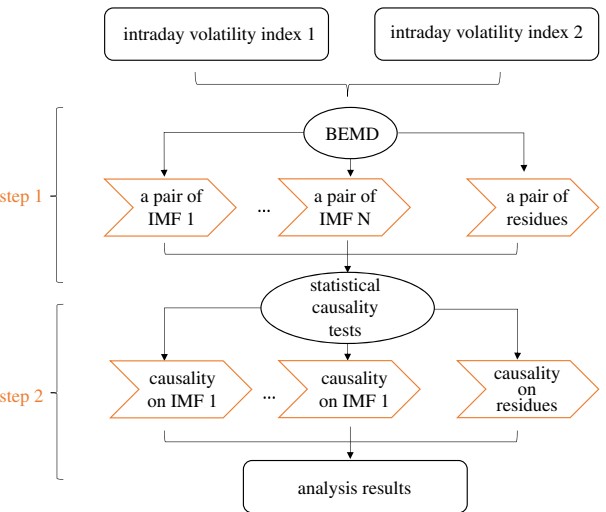

**Figure 3.** Diagram of the study. The methodology includes two steps, the bivariate empirical mode decomposition (BEMD) and the linear and nonlinear statistical causality tests.

14.30 (CET) for ECB announcement days and the time period 11.45 to 12.45 (GMT) for BoE announcement days, and adjust the time series such that they are both expressed in the same time zone.

From the logarithmic returns, we estimate the intraday volatility using a non-parametric estimator, the annualized *realized volatility*, constructed as the square root of the averaged squared returns multiplied by an annualizing factor, which is

$$\sigma_{t+i\Delta} = \sqrt{252n}\sqrt{\frac{1}{5}\sum_{k=1}^{5} r_{t+(i-k)\Delta}^2}. \tag{4.2}$$

The first two terms in equation (4.2) are the annualizing factor, which assumes 252 trading days per year, whereas $n = 509$ is the number of returns per trading day. The rest is a running average across the last $k = 1, \ldots, 5$ intraday squared returns. The stationarity of these time series have been tested using the augmented Dickey–Fuller (ADF) test.

According to [20], a good sampling frequency $\Delta$ that reduces the bias but maintains the accuracy of the realized volatility measurement is extremely important to avoid any distortions caused by market microstructure frictions. Authors such as Andersen *et al.* and Bandi & Russell [21,22] use 5 min sampling frequency. In this study, we decide to use all data at our disposal, maintaining a sampling frequency $\Delta$ of 1 min. We, however, reduce microstructural noise by using of the moving average of 5 min in equation (4.2), that smooths out irregularities and improves the accuracy of the realized volatility estimation.

From equation (4.2), we average the realized volatility values over the monetary policy announcement days. In this way, we obtain two couples of averaged realized volatility series—one sampled around the ECB announcements and the other around the BoE announcements—which are used as input signals for the causality investigation described in §4.2.

## 4.2. Bivariate empirical mode decomposition and transfer entropy

In order to investigate the interrelation between the Eurozone stock market and the UK stock market, we use linear and nonlinear causality measures combined with the BEMD technique originally proposed by [12]. The framework of the method is presented in figure 3.

The model includes two steps. First, the BEMD decomposes the time series into pairs of bands at different timescales. Second, two different approaches to transfer entropy are applied to the pairs of the IMFs and residues. The first approach is linear and involves the application of the Granger–Geweke causality test, which uses vector auto-regressive techniques to measure the predictability of the time series. The second approach is nonlinear and requires the estimation of the transfer entropy, which employs conditional mutual information to detect the statistical causality between the series. We believe there are two main advantages in carrying out the analysis on each IMF obtained through

the BEMD decomposition. First, it allows to examine the impact of news announcements and its statistical significance on volatility fluctuations expressed at different frequencies—short-term, medium-term or long-term. Second, it has obvious advantages in processing non-stationary and nonlinear data [13]. For the sake of completeness, we report results for both the original time series as well as the IMFs and residue obtained through the BEMD decomposition.

We implement the Bivariate EMD using the Rlibeemd package[7] in R, while we use the PyCausality package [16][8] in Python to compute linear and nonlinear transfer entropy. Results are validated using both $p$-value and $Z$-score computed with a non-parametric approach that compares results with null-hypothesis non-causal transfer entropy values, $TE_s^{shuffle}$, obtained from time series where the time entries are shuffled independently and therefore any statistical causality based on time-dependence is eliminated.

## 4.3. Non-parametric statistical validation

In order to validate results, we use the $p$-value and $Z$-score as statistical measures of significance. Following [16], we adopt a non-parametric approach comparing results with null-hypothesis non-causal transfer entropy values, $TE_s^{shuffle}$, obtained from time series where the time entries are shuffled independently and therefore any statistical causality based on time-dependence is eliminated. We estimate the significance by computing $N_{shuffles} = 1000$ shuffled values and calculating the $p$-value defined as the proportion of shuffled series which have larger transfer entropy values than the original result,

$$p := \frac{1}{N_{shuffles}} \sum_{s=1}^{N_{shuffles}} O(TE_s^{shuffle} - TE),$$ (4.3)

where $O(\cdot)$ is the step function.

The $Z$-score is instead measured as the distance between the computed transfer entropy (TE) and the average transfer entropy computed from shuffled data ($\mu_{shuffle}$), standardized by the shuffled standard deviation ($\sigma_{shuffle}$)

$$Z := \frac{TE - \mu_{shuffle}}{\sigma_{shuffle}}.$$ (4.4)

We compute the $p$-value and $Z$-score in equation (4.3) and (4.4) for both the linear and nonlinear case.

Data accessibility. This article has no additional data.

Authors' contributions. V.V.: conceptualization, data curation, investigation, methodology, writing—original draft, writing—review and editing; T.A.: conceptualization, formal analysis, funding acquisition, investigation, methodology, project administration, resources, software, supervision, validation, writing—original draft, writing—review and editing.

All authors gave final approval for publication and agreed to be held accountable for the work performed therein.

Competing interests. We declare we have no competing interests.

Funding. T.A. acknowledges partial support from ESRC (ES/K002309/1), EPSRC (EP/P031730/1) and EC (H2020-ICT-2018-2 825215).

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
