## [Peer Review File · Royal Society Open Science]

Review History

RSOS-211342.R0 (Original submission)

Review form: Reviewer 1 (Dror Kenett)

Is the manuscript scientifically sound in its present form?

Yes

Are the interpretations and conclusions justified by the results?

Yes

Is the language acceptable?

Yes

Do you have any ethical concerns with this paper?

No

Have you any concerns about statistical analyses in this paper?

No

Recommendation?

Accept with minor revision (please list in comments)

Comments to the Author(s)

The authors present an interesting analysis of the complement of the UK and EU stock market, using information based measures. I find the paper interesting, and only have a few comments:

1. In the introduction, the first sentence states: "The key role of central banks has been to conduct monetary policy to achieve price stability (low and stable inflation) and help manage economic fluctuations." That is a complicated statement, and I would suggest reconsidering. For example, the U.S. Fed defines its mission to be monetary policy and full labor market participation, and this is possibly shared by other central banks. Also, the financial stability program in some central banks existed even before the 2008 crisis.
2. Further down in the discussion regarding the impact central banks have over financial markets, I would suggest considering some of the recent work that has come out examining the interventions of central banks on financial markets during the pandemic.
3. There have been quite a few papers that have come out over the past few years studying the affects of central banks announcements on stock market movements, and I would encourage the authors to look for some more references from finance journals. Moreover, there are papers that have looked at the impact of the time in the day when significant announcements are made, which would be worth reviewing and discussing. For example, the 2020 JFE paper by Julio Crego investigating the impact of news announcements in the oil industry.
4. Regarding the methodology, it would be helpful to see either in the main text or in an appendix some robustness checks comparing the proposed methodology to others. Also, some additional references to the use in other papers of transfer entropy would be useful. For example, see Junior, Leonidas Sandoval, Asher Mullokandov, and Dror Y. Kenett. "Dependency relations among international stock market indices." *Journal of Risk and Financial Management* 8, no. 2 (2015): 227-265.

Review form: Reviewer 2 (Tomas Scagliarini)

Is the manuscript scientifically sound in its present form?

Yes

Are the interpretations and conclusions justified by the results?

Yes

Is the language acceptable?

Yes

Do you have any ethical concerns with this paper?

No

Have you any concerns about statistical analyses in this paper?

Yes

Recommendation?

Accept with minor revision (please list in comments)

Comments to the Author(s)

The manuscript of Volta is a study of the impact of the central bank's announcements on price volatility and assesses their impact on market indices through a causality analysis. The paper is mostly sound and provides a useful contribution to the literature.

However, I have some minor concerns about the analysis:

1. In Section 4, in the data description paragraph is written that in the original dataset Time is reported in "local time zone". Later, you said that you computed the logarithmic returns from 8:00 to 16:00 (GMT). If FTSE100 and EuroStoxx50 are in different time zones, as I assume, I think it should be explicitly written in the beginning of the section how you aligned the two price series, as this would be a crucial preprocessing step.
2. It is not clear what are the advantages of using BEMD technique over using the original time series. Is it just to have an analysis over multiple time scales or helps to address other issues like non stationarity?
3. Both transfer entropy and granger causality assume stationarity of time series. Have you tested this hypothesis with ADF test or other tools?

Decision letter (RSOS-211342.R0)

Dear Professor Aste

On behalf of the Editors, we are pleased to inform you that your Manuscript RSOS-211342 "Causal coupling between European and UK markets triggered by announcements of monetary policy decisions" has been accepted for publication in Royal Society Open Science subject to minor revision in accordance with the referees' reports. Please find the referees' comments along with any feedback from the Editors below my signature.

Please submit your revised manuscript and required files (see below) no later than 7 days from today's (ie 24-Jan-2022) date. Note: the ScholarOne system will 'lock' if submission of the revision is attempted 7 or more days after the deadline. If you do not think you will be able to meet this deadline please contact the editorial office immediately.

on behalf of Professor Andreas Kyprianou (Associate Editor) and Marta Kwiatkowska (Subject Editor)
openscience@royalsociety.org

Associate Editor Comments to Author (Professor Andreas Kyprianou):

Comments to the Author:

Both referees seem happy with the article, but request a few minor corrections before moving to publication.

Reviewer comments to Author:

Reviewer: 1

Comments to the Author(s)

The authors present an interesting analysis of the complement of the UK and EU stock market, using information based measures. I find the paper interesting, and only have a few comments:

1. In the introduction, the first sentence states: "The key role of central banks has been to conduct monetary policy to achieve price stability (low and stable inflation) and help manage economic fluctuations." That is a complicated statement, and I would suggest reconsidering. For example, the U.S. Fed defines its mission to be monetary policy and full labor market participation, and this is possibly shared by other central banks. Also, the financial stability program in some central banks existed even before the 2008 crisis.
2. Further down in the discussion regarding the impact central banks have over financial markets, I would suggest considering some of the recent work that has come out examining the interventions of central banks on financial markets during the pandemic.
3. There have been quite a few papers that have come out over the past few years studying the affects of central banks announcements on stock market movements, and I would encourage the authors to look for some more references from finance journals. Moreover, there are papers that have looked at the impact of the time in the day when significant announcements are made, which would be worth reviewing and discussing. For example, the 2020 JFE paper by Julio Crego investigating the impact of news announcements in the oil industry.
4. Regarding the methodology, it would be helpful to see either in the main text or in an appendix some robustness checks comparing the proposed methodology to others. Also, some additional references to the use in other papers of transfer entropy would be useful. For example, see Junior, Leonidas Sandoval, Asher Mullokandov, and Dror Y. Kenett. "Dependency relations among international stock market indices." *Journal of Risk and Financial Management* 8, no. 2 (2015): 227-265.

Reviewer: 2

Comments to the Author(s)

The manuscript of Volta is a study of the impact of the central bank's announcements on price volatility and assesses their impact on market indices through a causality analysis

The paper is mostly sound and provides a useful contribution to the literature.

However, I have some minor concerns about the analysis:

1. In Section 4, in the data description paragraph is written that in the original dataset Time is reported in "local time zone". Later, you said that you computed the logarithmic returns from 8:00 to 16:00 (GMT). If FTSE100 and EuroStoxx50 are in different time zones, as I assume, I think it should be explicitly written in the beginning of the section how you aligned the two price series, as this would be a crucial preprocessing step.
2. It is not clear what are the advantages of using BEMD technique over using the original time series. Is it just to have an analysis over multiple time scales or helps to address other issues like non stationarity?
3. Both transfer entropy and granger causality assume stationarity of time series. Have you tested this hypothesis with ADF test or other tools?

===PREPARING YOUR MANUSCRIPT===

one version should clearly identify all the changes that have been made (for instance, in coloured highlight, in bold text, or tracked changes);

===PREPARING YOUR REVISION IN SCHOLARONE===

-- If you are requesting an article processing charge waiver, you must select the relevant waiver option (if requesting a discretionary waiver, the form should have been uploaded, see 'File upload' above).

-- If you have uploaded any electronic supplementary (ESM) files, please ensure you follow the guidance at <https://royalsociety.org/journals/authors/author-guidelines/#supplementary-material> to include a suitable title and informative caption. An example of appropriate titling and captioning may be found at https://figshare.com/articles/Table_S2_from_Is_there_a_trade-off_between_peak_performance_and_performance_breadth_across_temperatures_for_aerobic_scope_in_teleost_fishes_/3843624.

Author's Response to Decision Letter for (RSOS-211342.R0)

See Appendix A.

Decision letter (RSOS-211342.R1)

Dear Professor Aste,

I am pleased to inform you that your manuscript entitled "Causal coupling between European and UK markets triggered by announcements of monetary policy decisions" is now accepted for publication in Royal Society Open Science.

on behalf of Professor Andreas Kyprianou (Associate Editor) and Marta Kwiatkowska (Subject Editor)
openscience@royalsociety.org

Appendix A

UNIVERSITY COLLEGE LONDON

Tomaso Aste

Professor of Complexity Science
Head, Financial Computing and Analytics Group
Director of UCL Centre for Blockchain Technologies
Director, MSc Financial Risk Management Programme
Vice-Director, UK Centre for Doctoral Training in
Financial Computing & Analytics
Board member, LSE Systemic Risk Centre
Member, UCL Centre for the Study of Decision-Making
Uncertainty
http://www.cs.ucl.ac.uk/staff/tomaso_aste/
E: t.aste@ucl.ac.uk
P: +44 203 108 7103
F: +44 20 7387 1397

August 16, 2021

Dear Editor,

Please find enclosed a paper entitled “Causal coupling between European and UK markets triggered by announcements of monetary policy decisions” that we submit for publication in Open Science as a regular research article.

The paper concerns a topic of general interest answering the question concerning *how decisions from regulators affect markets*. To provide a quantitative answer we use state-of-the-art tools which quantify causality as a flow of information. The results are clean and for the first time show that indeed in the minutes after central banks announcements the reference market has a surge in volatility that then spills over into other markets with a measurable causality effect that we quantify at different time scales.

We believe this paper will be of interest to Open Science readers and will have an impact on both academia, industry, and regulators.

All submitted material is original from the authors, it hasn't been published or submitted elsewhere.

With kind regards,
Tomaso Aste & Vittoria Volta
Sincerely,

Tomaso Aste